# Severe COVID-19 infection: An institutional review and literature overview

Ogheneyoma Akpoviroro[1]*, Nathan Kyle Sauers[2], Queeneth Uwandu[1], Myriam Castagne[3], Oghenetejiro Princess Akpoviroro[4], Sara Humayun[1], Wasique Mirza[1], Jameson Woodard[1]

1 Department of Internal Medicine, Geisinger Wyoming Valley Medical Center, Wilkes-Barre, Pennsylvania, United States of America, 2 Department of Engineering, Pennsylvania State University, State College, Pennsylvania, United States of America, 3 Clinical & Translational Science Institute, Boston University, Boston, Massachusetts, United States of America, 4 Department of Medicine, Mater Dei Hospital, Msida, Malta

* o.akpoviroro@gmail.com, oakpoviroro@geisinger.edu

## Abstract

### Background

Our study aimed to describe the group of severe COVID-19 patients at an institutional level, and determine factors associated with different outcomes.

### Methods

A retrospective chart review of patients admitted with severe acute hypoxic respiratory failure due to COVID-19 infection. Based on outcomes, we categorized 3 groups of severe COVID-19: (1) Favorable outcome: progressive care unit admission and discharge (2) Intermediate outcome: ICU care (3) Poor outcome: in-hospital mortality.

### Results

Eighty-nine patients met our inclusion criteria; 42.7% were female. The average age was 59.7 (standard deviation (SD):13.7). Most of the population were Caucasian (95.5%) and non-Hispanic (91.0%). Age, sex, race, and ethnicity were similar between outcome groups. Medicare and Medicaid patients accounted for 62.9%. The average BMI was 33.5 (SD:8.2). Moderate comorbidity was observed, with an average Charlson Comorbidity index (CCI) of 3.8 (SD:2.6). There were no differences in the average CCI between groups(p = 0.291). Many patients (67.4%) had hypertension, diabetes (42.7%) and chronic lung disease (32.6%). A statistical difference was found when chronic lung disease was evaluated; p = 0.002. The prevalence of chronic lung disease was 19.6%, 27.8%, and 40% in the favorable, intermediate, and poor outcome groups, respectively. Smoking history was associated with poor outcomes (p = 0.04). Only 7.9% were fully vaccinated. Almost half (46.1%) were intubated and mechanically ventilated. Patients spent an average of 12.1 days ventilated (SD:8.5), with an average of 6.0 days from admission to ventilation (SD:5.1). The intermediate group had a shorter average interval from admission to ventilator (77.2 hours, SD:67.6), than the poor group (212.8 hours, SD:126.8); (p = 0.001). The presence of bacterial

**Data Availability Statement:** The Data Availability Statement is under discussion and will be provided in a forthcoming update to this article.

**Funding:** The author received no specific funding for this work.

**Competing interests:** The authors have declared that no competing interests exist.

**Abbreviations:** ACE2, angiotensin converting enzyme 2; ACEi, angiotensin converting enzyme inhibitor; AKI, acute kidney injury; ALP, alkaline phosphatase; ALT, alanine aminotransferase; Ang I, angiotensin I; Ang II, angiotensin II; ARB, angiotensin receptor blocker; ARDS, acute respiratory distress syndrome; AST, aspartate aminotransferase; $AT_1R$, angiotensin II type 1 receptors; $AT_2R$, angiotensin II type 2 receptors; BMI, body mass index; BUN, blood urea nitrogen; CCI, Charlson comorbidity index; CHF, congestive heart failure; CKD, chronic kidney disease; CLD, chronic lung disease; COPD, chronic obstructive pulmonary disease; CRP, C-reactive protein; CSS, cytokine storm syndrome; DABK, des-arg-973-bradykinin; DM2, diabetes mellitus type 2; ESR, erythrocyte sedimentation rate; HTN, hypertension; KKS, kinin-kallikrein system; OSA, obstructive sleep apnea; PCU, progressive care unit; RAS, renin-angiotensin-aldosterone system; SD, standard deviation.

pneumonia was greatest in the intermediate group (72.2%), compared to the favorable group (17.4%), and the poor group (56%); this was significant (p<0.0001). In-hospital mortality was seen in 28.1%.

## Conclusion

Most patients were male, obese, had moderate-level comorbidity, a history of tobacco abuse, and government-funded insurance. Nearly 50% required mechanical ventilation, and about 28% died during hospitalization. Bacterial pneumonia was most prevalent in intubated groups. Patients who were intubated with a good outcome were intubated earlier during their hospital course, with an average difference of 135.6 hours. A history of cigarette smoking and chronic lung disease were associated with poor outcomes.

## Introduction

The worldwide COVID-19 pandemic significantly affected health systems, local communities and expected norms; this pandemic created a significant health resource scarcity in many geographic locations worldwide. Numerous descriptive studies have followed this pandemic as it continues, attempting to describe red flag patients that may require greater acuity of medical care.

Studies have shown that elevated inflammatory markers including inflammatory cytokines such as IL-6 [1], C-reactive protein (CRP), erythrocyte sedimentation rate (ESR), ferritin, markers of liver damage: alanine aminotransferase (ALT), aspartate aminotransferase (AST) and alkaline phosphatase (ALP), markers of kidney injury: blood urea nitrogen (BUN) and creatinine, amongst other markers of organ damage and inflammation, are associated with worse clinical course in COVID-19 infection [2,3]. Some studies have also found that certain laboratory tests, such as high albumin levels [4], may be a marker of less severe infection, while others such as leukopenia, may be associated with worse outcomes [1].

Multiple studies have also shown an association between severe COVID-19 infection including mortality, and the variables of older age and certain comorbidities, including cardiovascular disease, chronic respiratory disease, metabolic disorders including obesity and diabetes, and malignancies [2,3,5–9].

To determine the factors associated with poor outcomes, including intubation and in-hospital mortality in patients admitted with acute hypoxic respiratory failure and requiring high flow supplemental oxygen at admission at our institution, we performed this study. We aimed to describe the group of patients with severe COVID-19 infection, not only to ascertain factors associated with the development of severe COVID-19 infection at an institutional level, but also to describe factors associated with different outcome groups within the group of severe infection. COVID-19 is a novel viral infection that continues to persist as a long uphill battle of continuous research development of treatments and preventive measures. Patient care management requires the understanding of disease process and the risks associated with the disease.

## Materials and methods

### Design, setting, and patient population

A retrospective chart review study involving data collection from multiple Geisinger hospitals in Pennsylvania, USA. Geisinger Health System includes 10 major hospitals located in Central,

South-Central, North-Central and Northeast regions of the state of Pennsylvania, USA. These hospitals serve over 3 million residents in these regions. Only Geisinger Health System's data was used. We used the following inclusion criteria: admitted with acute hypoxic respiratory failure due to COVID-19 infection and requiring high flow nasal canula oxygen supplementation at admission, admission to the progressive care unit(PCU), age of 18–85 years, having received a 10-day course of high-dose dexamethasone (20mg for 5 days, followed by 10mg for a subsequent 5 days), and a 10-day course of remdesivir, and admission between 01/01/2021-12/31/2021. We defined severe COVID-19 infection as patients who required admission to the progressive care unit (PCU) of our institution. Based on outcomes, we categorized 3 groups of severe COVID-19: (1) Favorable outcome: this included patients who were admitted to the PCU, then discharged from the hospital, (2) Intermediate outcome: this group included patients who were initially admitted to the PCU but required escalation to an intensive cause unit (ICU) level of care and eventually discharged, and (3) Poor outcome: this group included patients with in-hospital mortality. ICU level of care was defined as requiring intubation and mechanical ventilation support. Some of the authors of this manuscript had access to identifiable patient information during data collection (Fig 1).

### Data collection and handling

Patients were identified using diagnosis on inpatient problem list between 01/01/2021-12/31/2021. Comorbidities were identified by ICD9 and ICD10 diagnosis codes, including tobacco use history, and required at least one reference to diagnosis on the inpatient problem list, or two outpatient diagnoses. Medication use was determined by orders within the Geisinger system and medication reconciliation during encounters. Charlson comorbidity index was calculated for each patient to determine comorbid burden.

### Statistical methods

ANOVA test was used for comparison between the 3 groups. A post-hoc Tukey's HSD test was conducted to compare the differences between 2 groups. IBM SPSS Statistics 26 was used for all analyses. Significance levels of $<0.05$ and $<0.0167$ were set for the ANOVA and Tukey's HSD tests respectively.

### Ethical approval

This study was approved by the institutional review board of Geisinger Northeast, Wilkes Barre, Pennsylvania as an exempt study, as defined by the U. S. Department of Health and Human Services Regulations for the Protection of Human Subjects [(45 CFR 46.104)]. This means that patient consent was not required for this retrospective study, with the requirement that only the investigators approved to participate in this study have access to identifiable patient data and information, and any data and results published that are associated with this study must be fully de-identified before publication. The institutional review board of Geisinger Northeast, Wilkes Barre, Pennsylvania, USA has approved the results of this study to be published.

## Results

### Demographics

The study population included 89 patients; 42.7% were female and 57.3% were male. A favorable outcome occurred in 51.7% (46/89) of patients with severe COVID-19 infection, 20.2% (18/89) had an intermediate outcome, and 28.1% (25/89) had a poor outcome. The average age

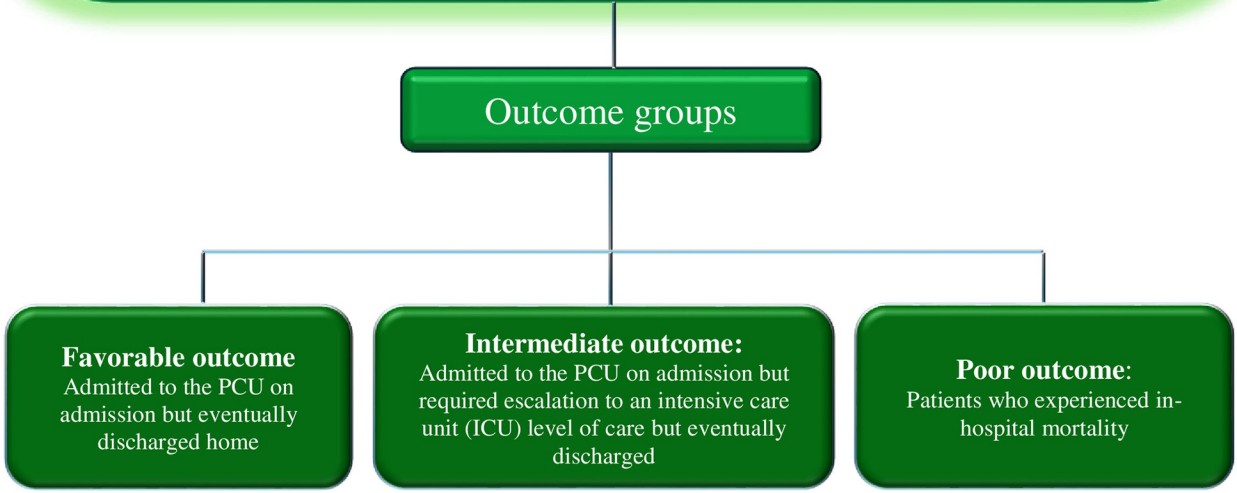

**Inclusion criteria:**

1) Admission diagnosis: acute hypoxic respiratory failure due to COVID-19 infection

2) High-flow nasal canula oxygen supplementation at admission

3) Admitted to the PCU (i.e. severe COVID-19)

4) Aged 18-85 years

5) Received 10 days of high-dose steroids (20mg dexamethasone for 5 days followed by 10mg dexamethasone for 5 days)

6) Received 10 days of remdesivir

Outcome groups

**Favorable outcome**
Admitted to the PCU on admission but eventually discharged home

**Intermediate outcome:**
Admitted to the PCU on admission but required escalation to an intensive care unit (ICU) level of care but eventually discharged

**Poor outcome**:
Patients who experienced in-hospital mortality

**Fig 1. Inclusion and eligibility.** PCU, progressive care unit; ICU level of care equates requiring intubation and mechanical ventilation support.

of the entire study population was 59.7 (standard deviation (SD):13.7). The average ages of the 3 outcome groups were 59.8(SD 13.1), 55.1(SD 14.6), and 62.8(SD 12.9) years in the favorable, intermediate, and poor groups, respectively.

Most of the entire study population were male (57.3%), with females constituting 42.7%. Female patients made up 39.1%, 61.1% and 36.0% of the favorable, intermediate, and poor outcome groups, respectively. Male patients made up 60.9%, 38.9% and 64.0% of the favorable, intermediate, and poor outcome groups, respectively.

Of the total study population, Caucasian patients were 95.5%, and 4.5% were Black or African American. Caucasian patients made up >90% of each outcome group Table 1.

**Table 1. Demographic data.**

| Variables | | Favorable n (%) | Intermediate n (%) | Poor n (%) | Total n (%) | P-value* | P-value** | P-value*** | P-value*^ |
|---|---|---|---|---|---|---|---|---|---|
| Sex | Female | 18 (39.1) | 11 (61.1) | 9 (36) | 38 (42.7) | 0.208 | 0.251 | 0.965 | 0.233 |
| | Male | 28 (60.9) | 7 (38.9) | 16 (64) | 51 (57.3) | | | | |
| | **Total** | 46 | 18 | 25 | 86 | | | | |
| Race | African American/Black | 2 (4.3) | 1 (5.6) | 1 (4) | 4 (4.5) | 0.970 | 0.977 | 0.998 | 0.969 |
| | Caucasian | 44 (95.7) | 17 (94.4) | 24 (96) | 85 (95.5) | | | | |
| | **Total** | **46** | **18** | **25** | **89** | | | | |
| **Ethnicity** | Hispanic | 1 (2.2) | 3 (16.7) | 1 (4) | 5 (5.6) | **0.002** | 0.002 | 0.945 | 0.011 |
| | Non-Hispanic | 45 (97.8) | 12 (66.7) | 24 (96.0) | 81 (91.0) | | | | |
| | Unknown | 0 | 3 (16.7) | 0 | 3 (3.4) | | | | |
| | **Total** | **46** | **18** | **25** | **89** | | | | |
| Age (years) | Mean | 59.8 | 55.1 | 62.8 | 59.7 | 0.189 | 0.431 | 0.642 | 0.162 |
| | SD | 13.1 | 14.6 | 12.9 | 13.7 | | | | |
| | IQR | 19.3 | 15 | 16 | 17 | | | | |
| Insurance | Commercial | 18 (39.1) | 6 (33.3) | 9 (36) | 33 (37.1) | 0.906 | 0.905 | 0.964 | 0.983 |
| | Medicaid/Medicare | 28 (60.9) | 12 (66.7) | 16 (64) | 46 (62.9) | | | | |
| | Self-paying | 0 | 0 | 0 | 0 | | | | |
| | **Total** | 46 | 18 | 25 | 89 | | | | |
| Vaccination | Y | 4(8.7) | 1 (5.6) | 2(8) | 7(7.9) | 0.806 | 0.793 | 0.994 | 0.870 |
| | N | 42(91.3) | 17(94.4) | 23(92) | 82(92.1) | | | | |
| | **Total** | 46 | 18 | 25 | 89 | | | | |

* Three-group comparison

** Favorable versus intermediate

*** Favorable versus poor

*^Intermediate versus poor; SD, standard deviation; IQR, interquartile range.

Most of the total population were non-Hispanic at 91.0%; 5.6% were Hispanic and 3.4% had no ethnicity data available. Non-Hispanic patients made up 97.8%, 66.7%, and 96.0% of the favorable, intermediate, and poor outcome groups, respectively.

There were no statistical differences in sex, when all outcome groups were compared (p = 0.208), or between intergroup comparisons Table 1. More Hispanic patients were present in the intermediate group (16.7%) than the favorable (2.2%), or poor (4%) outcome groups, and these differences were statistically significant(p = 0.002) Table 1. Only 7.9% of the entire population were fully vaccinated. Of those vaccinated, 57.1% were in the favorable group, 14.3% in the intermediate group, and 28.6% in the poor group Table 1.

## Insurance

Medicare and Medicaid patients accounted for 62.9% of the study population, with private insurance in 37.1%. All outcome groups similarly had around 60% of Medicare/Medicaid insurance, and there were no differences between the outcome groups(p = 0.906) Table 1.

## Comorbidities

The average body mass index (BMI) was 33.5 (SD:8.2). Patients in all outcome groups were categorized as class I obese, with an average BMI in the favorable, intermediate, and poor outcome groups of 32.3 (SD 6.8), 34.9 (SD 10.7), and 34.7 (SD 8.0), respectively. No differences existed between outcome groups (p = 0.367) Table 2.

**Table 2. Comorbidities.**

| Variables | | Favorable n (%) | Intermediate n (%) | Poor n (%) | Total n (%) | P-value* | P-value** | P-value*** | P-value*^ |
|---|---|---|---|---|---|---|---|---|---|
| BMI | Mean | 32.3 | 34.9 | 34.7 | 33.5 | 0.367 | 0.499 | 0.468 | 0.998 |
| | SD | 6.8 | 10.7 | 8.0 | 8.2 | | | | |
| CCI | Mean | 3.5 | 3.5 | 4.5 | 3.8 | 0.291 | 0.998 | 0.304 | 0.429 |
| | SD | 2.6 | 2.5 | 2.4 | 2.6 | | | | |
| CKD | Yes | 4 (8.7) | 2 (11.1) | 0 (0) | 10 (11.2) | 0.656 | 0.960 | 0.630 | 0.874 |
| | No | 42 (91.3) | 16 (88.9) | 25 (100) | 79 (88.8) | | | | |
| | Total | 46 | 18 | 25 | 89 | | | | |
| DM2 | Yes | 20 (43.5) | 10 (55.6) | 8 (32.0) | 38 (42.7) | 0.309 | 0.657 | 0.622 | 0.280 |
| | No | 26 (56.5) | 8 (44.4) | 17 (68.0) | 51 (57.3) | | | | |
| | Total | 46 | 18 | 25 | 89 | | | | |
| HTN | Yes | 30 (65.2) | 14 (77.8) | 16 (64.0) | 60 (67.4) | 0.582 | 0.608 | 0.994 | 0.616 |
| | No | 16 (34.8) | 4 (22.2) | 9 (36.0) | 29 (32.6) | | | | |
| | Total | 46 | 18 | 25 | 89 | | | | |
| CHF | Yes | 8 (17.4) | 2 (11.1) | 4 (16.0) | 14 (15.7) | 0.829 | 0.814 | 0.987 | 0.904 |
| | No | 38 (82.6) | 16 (88.9) | 21 (84.0) | 75 (84.3) | | | | |
| | Total | 46 | 18 | | | | | | |
| **CLD** | Yes | 9 (19.6) | 5 (27.8) | 10 (40.0) | 29 (32.6) | **0.002** | 0.783 | 0.001 | 0.054 |
| | No | 37 (80.4) | 13 (72.2) | 15 (60.0) | 60 (67.4) | | | | |
| | Total | 46 | 18 | 25 | 89 | | | | |
| COPD | Yes | 8 (17.4) | 2 (11.1) | 7 (28.0) | 17 (19.1) | 0.356 | 0.836 | 0.529 | 0.354 |
| | No | 38 (82.6) | 16 (88.9) | 18 (72.0) | 72 (80.9) | | | | |
| | Total | 46 | 18 | 25 | 89 | | | | |
| **OSA** | Yes | 4 (8.7) | 2 (11.1) | 9 (36.0) | 15 (16.9) | **0.009** | 0.969 | 0.009 | 0.354 |
| | No | 42 (91.3) | 16 (88.9) | 16 (64.0) | 74 (83.1) | | | | |
| | Total | 46 | 18 | 25 | 89 | | | | |
| **Ever Smoker** | Yes | 22 (47.8) | 8 (44.4) | 19 (76.0) | 49 (55.1) | **0.044** | 0.966 | 0.058 | 0.098 |
| | No | 24 (52.2) | 10 (55.6) | 6 (24.0) | 40 (44.9) | | | | |
| | Total | 46 | 18 | 25 | 89 | | | | |
| Former smoker | Yes | 18(81.8) | 8(100) | 14(73.7) | 40(81.6) | 0.285 | 0.499 | 0.783 | 0.254 |
| | No | 4(18.2) | 0 | 5(26.3) | 9(18.4) | | | | |
| | Total | 22 | 8 | 19 | 49 | | | | |

\* Three-group comparison

\*\* Favorable versus intermediate

\*\*\* Favorable versus poor

*^Intermediate versus poor; SD, standard deviation; IQR, interquartile range; BMI, body mass index; CCI, Charlson Comorbidity index; CKD, chronic kidney disease; DM2, type 2 diabetes mellitus; HTN, hypertension; CHF, congestive heart failure; CLD, chronic lung disease; COPD, chronic obstructive pulmonary disease; OSA, obstructive sleep apnea.

Moderate comorbidity was observed in the total population, with a mean Charlson Comorbidity index (CCI) of 3.8 (SD:2.6). The average CCI in the favorable, intermediate, and poor outcome groups were 3.5 (SD 2.6), 3.5 (SD 2.5), and 4.5 (SD 2.4), respectively(p = 0.291) Table 2.

The prevalence of chronic kidney disease (CKD) in the entire study population was 11.2%; the prevalence was 8.7%, 11.1%, and 0% in the favorable, intermediate, and poor outcome groups, respectively(p = 0.656).

The overall prevalence of type 2 diabetes (DM2) was 42.7%, and this was similar in the favorable (43.5%), intermediate (55.6%), and poor (32.0%) outcome groups(p = 0.309). Hypertension(HTN) was prevalent in 67.4% of the entire population, and occurred in 65.2%, 77.8%, and 64.0% of the favorable, intermediate, and poor outcome groups, respectively(p = 0.582).

Congestive heart failure (CHF) was prevalent in 15.7% of the entire study population and occurred at a similar rate in all outcome groups(p = 0.829).

Although BMI, CCI, CKD, DM2, HTN, or CHF did not show any significant differences between outcome groups Table 2, there was a noticeable difference between the outcome groups regarding chronic lung disease (CLD). The prevalence of CLD in the entire severe COVID-19 study population was 32.6%, with a stepwise increase in prevalence with increasing outcome severity. CLD was found in 19.6%, 27.8%, and 40% of the favorable, intermediate, and poor outcome groups, respectively. These findings were significant when all three groups were compared against each other (p = 0.002), and in all intergroup comparisons except favorable versus intermediate group Table 2.

Chronic obstructive pulmonary disease (COPD) occurred in 19.1% of the entire study population, and in 17.4%, 11.1% and 28.0% of the favorable, intermediate, and poor outcome groups, respectively(p = 0.356) Table 2. Obstructive sleep apnea (OSA) showed an upward trend with each subsequent outcome group (8.7%, 11.1%, 36.0%), and this was statistically significant(p = 0.009) in the 3-group comparison Table 2.

A smoking history was found in 55.1% of the entire study population, and in 47.8%, 44.4%, and 76.0% of the favorable, intermediate, and poor outcome groups, respectively. These findings were significant(p = 0.044) Table 2. We further evaluated the type of smoking history to determine if former smokers had a different outcome than current smokers. Former smokers were defined as those with a record of smoking cessation for at least 6 months per chart review; most patients who fell into this category had a record of having quit for greater than 1 year. Most smokers in all outcome groups were found to be former smokers, and there were no differences between the outcome groups(p = 0.285) Table 2.

## Hospital course

Patients were on average admitted for 22.1 days. The mean length of stay was 17.1, 23.6 and 21.3 days in the favorable, intermediate, and poor outcome groups, respectively(p = 0.013) Table 3.

Most patients(75.3%) needed non-invasive ventilation(NIV), and while more patients in the intermediate(88.9%) and poor(84.0%) groups needed NIV than the favorable group (65.2%), these differences were not statistically significant(p = 0.071) Table 3.

Almost half (44.9%) of the severe COVID-19 population were intubated and mechanically ventilated. All patients in the intermediate group where intubated and mechanically ventilation per definition, and no patients in the favorable group were intubated and mechanically ventilated. All but 3 patients in the poor group were intubated(88.0%) Table 3. The average interval to intubation was 144.6 hours (SD 122.0). Of those who were intubated, the interval from admission to intubation was 77.2 hours(SD 67.6) and 212.8 hours(SD 126.8) in the intermediate and poor groups, respectively(p = 0.001) Table 3. The average number of days intubated was 12.1 days, and this was similar between the intermediate(13.2, SD 9.7) and poor (10.6, SD 7.2) groups.

The prevalence of bacterial pneumonia was greatest in the intermediate group (72.2%) than the favorable (17.4%), or poor (56%) groups(p<0.0001). These differences were significant in all intergroup comparisons except between intermediate and poor (p = 0.455).

In-hospital mortality occurred in 28.1% (25/89), and these patients belonged to the poor outcome group Table 3.

**Table 3. Hospital course.**

| Variables | | Favorable n (%) | Intermediate n (%) | Poor n (%) | Total n (%) | P-value* | P-value** | P-value*** | P-value*^ |
|---|---|---|---|---|---|---|---|---|---|
| Length of stay (days) | Mean | 17.1 | 23.6 | 21.3 | 22.1 | 0.013 | 0.018 | 0.117 | 0.644 |
| | SD | 7.2 | 10.8 | 7.7 | 9.56 | | | | |
| Non-invasive ventilation | Yes | 30(65.2) | 16(88.9) | 21(84) | 67(75.3) | 0.071 | 0.118 | 0.184 | 0.927 |
| | No | 16(34.8) | 2(11.1) | 4(16) | 22(24.7) | | | | |
| | Total | 46 | 18 | 25 | 89 | | | | |
| Invasive ventilation | Yes | 0(0) | 18(100) | 22(88.0) | 40(44.9) | <0.001 | <0.001 | <0.001 | 0.060 |
| | No | 46(100) | 0(0) | 3(12.0) | 49(55.1) | | | | |
| | Total | 46(100) | 18(100) | 0 | 89 | | | | |
| Interval from admission to mechanical ventilation (hours) | Mean | n/a | 77.2 | 212.8 | 144.6 | n/a | n/a | n/a | |
| | SD | n/a | 67.6 | 126.8 | 122.0 | | | | **0.001** |
| Number of days on mechanical ventilator | Mean | n/a | 13.2 | 10.6 | 12.1 | n/a | n/a | n/a | 0.474 |
| | SD | n/a | 9.7 | 7.2 | 8.5 | | | | |
| Bacterial pneumonia | Yes | 8(17.4) | 13(72.2) | 14(56.0) | 35(39.3) | <0.001 | <0.001 | 0.002 | 0.455 |
| | No | 38(82.6) | 5(27.8) | 11(44.0) | 54(60.7) | | | | |
| | Total | 46 | 18 | 25 | 89 | | | | |
| In-hospital mortality | Yes | 0(0) | 0(0) | 23(92.0) | 23(25.8) | n/a | n/a | n/a | n/a |
| | No | 46(100) | 18(100) | 2(8.0) | 66(74.2) | | | | |
| | Total | 46 | 18 | 25 | 89 | | | | |

* Three-group comparison

** Favorable versus intermediate

*** Favorable versus poor

*^Intermediate versus poor.

## Laboratory values

**Liver function panel.** The average albumin in the entire study population was 3.0(SD 0.4); the favorable outcome group had the lowest average albumin levels during admission (2.8, SD: 0.3), compared with the intermediate (3.2, SD:0.3) and the poor (3.0, SD: 0.7) outcome groups(p = 0.001) Table 4. The average population AST was 56.1(SD 53.7) and trended up with each subsequent outcome group. Average AST was 43.4 (SD 16.1), 49.0(SD 20.7), and 84.7(SD 90.7) in the favorable, intermediate, and poor outcome groups, respectively(p = 0.006) Table 4. The average ALT in the total population was 68.5 (SD 127.0) and was not statistically different between outcome groups, even though it trended up with each subsequent outcome group(p = 0.197) Table 4. The average total population bilirubin was 0.5(SD 0.2), and this was highest in the poor outcome group (0.5, SD:0.2) than the favorable (0.4, SD:0.2), or the intermediate (0.4, SD:0.2) groups(p = 0.042) Table 4.

**Markers of inflammation.** Average CRP in the total study population was 69.5 (SD 50.2), and was greater in the intermediate (111.5, SD 69.8) and poor (65.7, SD 31.9) groups than the favorable (56.0, SD 39.6) group(p<0.001) Table 4.

The total population average lactate was 2.2(SD 2.3). Average lactate was significantly elevated in the poor outcome group (3.4, SD 3.8), than the favorable (1.8, SD 0.6), or intermediate (1.6, SD 0.3) groups(p = 0.02) Table 4. The average total population lymphocyte count was

**Table 4. Laboratory tests.**

| Variable | | Favorable n (%) | Intermediate n (%) | Poor n (%) | Total n (%) | P-value* | P-value** | P-value*** | P-value*^ |
|---|---|---|---|---|---|---|---|---|---|
| Average albumin (g/dL) | Mean | 2.8 | 3.2 | 3.0 | 3.0 | **0.001** | 0.001 | 0.144 | 0.131 |
| | SD | 0.3 | 0.3 | 0.4 | 0.4 | | | | |
| Average AST (U/L) | Mean | 43.4 | 49.0 | 84.7 | 56.1 | **0.006** | 0.918 | 0.005 | 0.068 |
| | SD | 16.1 | 20.7 | 90.7 | 53.7 | | | | |
| Average ALT (U/L) | Mean | 54.5 | 50.4 | 107.3 | 68.5 | 0.197 | 0.992 | 0.216 | 0.397 |
| | SD | 34.8 | 33.6 | 227.2 | 127.0 | | | | |
| Average bilirubin (mg/dL) | Mean | 0.4 | 0.4 | 0.5 | 0.5 | 0.042 | 0.616 | 0.127 | 0.045 |
| | SD | 0.2 | 0.2 | 0.2 | 0.2 | | | | |
| Average CRP (mg/L) | Mean | 56.0 | 111.5 | 65.7 | 69.5 | **<0.001** | <0.001 | 0.678 | 0.006 |
| | SD | 39.6 | 69.8 | 31.9 | 50.2 | | | | |
| Average Ferritin (ng/mL) | Mean | 1398.6 | 917.3 | 1241.1 | 1259.3 | 0.356 | 0.324 | 0.853 | 0.655 |
| | SD | 1349.5 | 704.0 | 1020.3 | 1172.5 | | | | |
| Average lactate (mmol/L) | Mean | 1.8 | 1.6 | 3.4 | 2.2 | **0.02** | 0.94 | 0.033 | 0.046 |
| | SD | 0.6 | 0.3 | 3.8 | 2.3 | | | | |
| Average lymphocyte(K/uL) | Mean | 0.9 | 1.1 | 0.8 | 0.9 | 0.381 | 0.680 | 0.713 | 0.347 |
| | SD | 0.5 | 0.6 | 0.4 | 0.5 | | | | |
| Average WBC (K/uL) | Mean | 11.8 | 13.4 | 14.3 | 12.8 | 0.029 | 0.314 | 0.027 | 0.704 |
| | SD | 3.7 | 3.6 | 4.4 | 4.0 | | | | |
| Average procalcitonin (ng/mL) | Mean | 0.3 | 1.3 | 0.4 | 0.5 | 0.173 | 0.167 | 0.994 | 0.262 |
| | SD | 0.5 | 4.0 | 0.6 | 1.9 | | | | |
| AKI | Yes | 2 (4.3) | 1 (5.6) | 5 (20) | 8 (9) | 0.076 | 0.987 | 0.072 | 0.229 |
| | No | 44 (95.7) | 17 (94.4) | 20 (80) | 81 (91) | | | | |
| | Total | 46 | 18 | 25 | 89 | | | | |

* Three-group comparison

** Favorable versus intermediate

*** Favorable versus poor

*^Intermediate versus poor.

0.9K/uL(SD 0.5) and was similar between outcome groups(p = 0.381) Table 4. Although there were no outcome differences in the average lymphocyte count, there was an uptrend in the total WBC count, and the poor group had a significantly higher average white blood cell count (WBC)(14.3, SD 4.4) than either the favorable (11.8, SD 3.7), or the intermediate (13.4, SD 3.6) groups(p = 0.029) Table 4.

As expected, the average procalcitonin level was elevated in the total study population(0.5, SD 1.9); however, there were no differences between the outcome groups(p = 0.173) Table 4.

**Renal function.** We further evaluated the presence of acute kidney injury (AKI). There was an insignificant upward trend in the prevalence of AKI in the outcome groups. While 4.3% in the favorable outcome group had AKI, 5.6% and 20% in the intermediate and poor outcome groups, respectively, had AKI(p = 0.076) Table 4.

## Respiratory parameters

**Patrial pressure of carbon dioxide and oxygen.** The average total population partial pressure of arterial carbon dioxide (PaCO2) was 41.4, and PaCO2 showed an uptrend with each subsequent outcome group(p<0.001) Table 5. Similarly, the maximum PaCO2 was above the

**Table 5. Respiratory parameters.**

| Variable | | Favorable n (%) | Intermediate n (%) | Poor n (%) | Total n (%) | P-value* | P-value** | P-value*** | P-value*^ |
|---|---|---|---|---|---|---|---|---|---|
| Average PaCO2(mmHg) | Mean | 35.4 | 39.9 | 46.4 | 41.4 | <0.001 | 0.313 | <0.001 | 0.007 |
| | SD | 6.4 | 8.5 | 14.4 | 11.6 | | | | |
| Maximum PaCO2(mmHg) | Mean | 36.5 | 52.5 | 64.8 | 52.3 | <0.001 | 0.015 | <0.001 | 0.004 |
| | SD | 7.4 | 12.2 | 30.0 | 23.2 | | | | |
| Minimum PaCO2(mmHg) | Mean | 34.3 | 25.2 | 32.0 | 31.5 | <0.001 | <0.001 | 0.758 | <0.001 |
| | SD | 6.0 | 4.3 | 7.7 | 6.9 | | | | |
| Average PaO2(mmHg) | Mean | 76.0 | 111.8 | 83.0 | 88.7 | <0.001 | <0.001 | 0.23 | 0.001 |
| | SD | 19.4 | 24.5 | 18.3 | 24.7 | | | | |
| Maximum PaO2(mmHg) | Mean | 83.9 | 227.6 | 132.8 | 141.7 | <0.001 | <0.001 | 0.001 | <0.001 |
| | SD | 27.4 | 71.8 | 55.5 | 77.5 | | | | |
| Minimum PaO2(mmHg) | Mean | 69.1 | 60.4 | 52.5 | 61.0 | <0.001 | 0.128 | <0.001 | 0.143 |
| | SD | 190 | 9.5 | 11.4 | 15.8 | | | | |
| Average pH | Mean | 7.45 | 7.40 | 7.36 | 7.40 | <0.001 | 0.001 | <0.001 | 0.009 |
| | SD | 0.042 | 0.035 | 0.061 | 0.060 | | | | |
| Maximum pH | Mean | 7.46 | 7.51 | 7.46 | 7.47 | 0.001 | 0.002 | 0.998 | 0.004 |
| | SD | 0.043 | 0.038 | 0.043 | 0.04 | | | | |
| Minimum pH | Mean | 7.44 | 7.27 | 7.25 | 7.33 | <0.001 | <0.001 | <0.001 | 0.205 |
| | SD | 0.047 | 0.076 | 0.14 | 0.13 | | | | |
| Respiratory rate on admission(bpm) | Mean | 23.8 | 27.8 | 23.3 | 24.5 | 0.049 | 0.069 | 0.944 | 0.064 |
| | SD | 5.9 | 7.9 | 5.2 | 6.5 | | | | |
| Average respiratory rate(bpm) | Mean | 22.2 | 21.2 | 24.0 | 22.5 | 0.008 | 0.478 | 0.044 | 0.01 |
| | SD | 2.7 | 2.4 | 3.7 | 3.2 | | | | |
| SpO2 on admission(%) | Mean | 89.0 | 86.0 | 89.1 | 88.4 | 0.402 | 0.409 | 1.000 | 0.476 |
| | SD | 7.9 | 10.1 | 8.0 | 8.5 | | | | |
| Minimum SpO2(%) | Mean | 76.5 | 70.8 | 57.0 | 69.9 | <0.001 | 0.451 | <0.001 | 0.024 |
| | SD | 13.2 | 18.6 | 19.8 | 18.5 | | | | |
| Average SpO2(%) | Mean | 93.5 | 94.6 | 92.2 | 93.4 | <0.001 | 0.009 | 0.004 | <0.001 |
| | SD | 1.0 | 1.2 | 1.9 | 1.6 | | | | |

* Three-group comparison

** Favorable versus intermediate

*** Favorable versus poor

*^Intermediate versus poor; mmHg, millimeters of mercury; bmp, breaths per minute.

upper limit of normal in the intermediate and poor groups, being greatest in the poor group (64.8, SD:30.0), than the intermediate (52.5, SD 12.2) or favorable (36.5, SD 7.4) groups (p<0.001) Table 5.

The minimum PaCO2 was lowest in the intermediate, followed by the poor, and then the favorable groups(p<0.001) Table 5.

The average partial pressure of arterial oxygen (PaO2) of the entire population was 88.7mmHg(SD24.7). Average PaO2 was highest in the intermediate group(111.8, SD 24.5) than the favorable (76.0, SD 19.4) or poor (83.0, SD 18.3) groups(p<0.001) Table 5. Similarly, the maximum PaO2 was greatest in the intermediate group, followed by the poor and then the favorable groups(p<0.001) Table 5. The minimum PaO2 was lowest in the poor group and highest in the favorable group(p<0.001) Table 5.

**pH.**   Average total population pH was 7.40(SD 0.06). The average pH of each outcome group was within normal limits. While the favorable outcome group had a more alkalotic average pH (7.45, SD 0.042), the intermediate (7.40, SD:0.035) and poor (7.36, SD:0.061) groups had slightly more acidic average pH(p<0.001) Table 4. The average minimum pH was 7.33, and it was most acidotic in the poor group(7.25, SD 0.14) than the intermediate(7.27, SD 0.076) or favorable(7.44, SD:0.047) groups(p<0.001) Table 5. The average maximum pH was greatest in the intermediate group than the other groups(p<0.001) Table 5.

**Respiratory rate and oxygen saturation.**   The respiratory rate (RR) on admission was greatest in the intermediate group (27.8, SD 7.9), compared to the favorable (23.8, SD 5.9) and poor (23.3, SD 5.2) outcome groups(p = 0.049). The average RR of the entire study population was 22.5(SD 3.7). The average RR was greatest in the poor(24.0, SD 3.7), than the favorable (22.2, SD 2.7) or intermediate (21.2, SD 2.4) groups(p = 0.008) Table 5. The average oxygen saturation(SpO2) on admission was 88.4% and the average values were similar across all outcome groups(p = 0.402) Table 5. The average SpO2 of the entire study population during admission was 93.4%(SD 1.6); the poor outcome group had the lowest average(92.2% SD 1.9) and minimum(57.0% SD 19.8) SpO2 values than the other outcome groups(p<0.001) Table 5. The average maximum SpO2 of the entire population was 99.9%, and no differences existed between the favorable(99.9% SD 0.4), intermediate(100% SD 0.0) and poor(99.8% SD 0.7) outcome groups(p = 0.308).

**Medications during admission.**   Approximately 4–5% of patients in each outcome group received hydroxychloroquine during admission(p = 0.970) Table 6 in S1 Table. Similarly, approximately 30% of patients in each subgroup received tocilizumab(p = 0.960) Table 6 in S1 Table. More patients in the poor outcome group received baricitinib(28%) than in the intermediate(5.6%) or poor(4.3%) groups(p = 0.007) Table 6 in S1 Table. There were no differences in the exposure to azithromycin in the three outcome groups(p = 0.268); all patients in all outcome groups received remdesivir for 10 days during admission. Similarly, all patients received dexamethasone during admission, and there were no differences in prior-to-admission receipt of dexamethasone between groups(p = 0.562) Table 6 in S1 Table. Patients in the poor outcome group had a longer average number of days on dexamethasone(16.6 days SD 4.3), than the intermediate(12.9 days SD 4.0) or the favorable (13.3 days SD 3.5) outcome groups (p = 0.002). Although the poor outcome group had the lowest proportion of patients on angiotensin receptor blockers(ARBs)(4%) and angiotensin converting enzyme inhibitors(ACEi) (12%) during admission, compared with the intermediate group(11.1% and 27.8%, respectively) and the favorable(8,7% and 23.9%, respectively) groups, these differences were not significant for either ARB exposure(p = 0.971), or ACEi exposure(p = 0.631) Table 6 in S1 Table. Thirty percent of the study population were on statin medications during admission, and more patients in the poor(48%) than the intermediate(27.8%) or favorable(28.3%) outcome groups were on statin medications during admission(p = 0.209) Table 6 in S1 Table.

## Discussion

### Severe COVID-19 and sex

Our study showed that several factors were associated with poor outcomes in severe COVID-19 infection, including intubation and in-hospital mortality. Similar to other studies [6,10–14], men were overrepresented in our study population, making up close to 60%. Similarly, a greater proportion of male patients experienced in-hospital mortality than did female patients. Studies have suggested that the overrepresentation of COVID-19 infection and poor outcomes in male patients may be associated with biological factors, and gender-specific behaviors such as poorer baseline health that might be associated with poor lifestyle choices that permit a

greater comorbid burden [15]. Such a theory is supported by studies that have shown a greater incidence of alcohol-attributable deaths and disease burden in men [16], greater participation in illicit drug use than females [17], and greater tobacco consumption [17,18]; in the resultant comorbidities such as chronic lung disease and cardiovascular disease have been linked to worse outcomes in COVID-19 [15]. Biological mechanisms that have been proposed to explain the worse prevalence and outcomes in male patients relate to how the SARS-CoV-2 virus enters the respiratory epithelium via transmembrane angiotensin converting enzyme-2(ACE2) also known as ACE2 receptor (ACE2r) [19]. Males have been shown to have increased androgen receptor expression compared to women [20]. Androgen receptors have been shown to have a positive correlation with ACE2 [21]. Increased ACE2 expression may increase susceptibility to SARS-CoV-2; it therefore follows that male patients may have a heightened risk of COVID-19 infection and more invasive disease than females [15,21]. Additionally, behavioral factors that increase ACE2 expression, specifically smoking [15], may also increase susceptibility to COVID-19 infection in male patients in an additive manner. Other biological mechanisms proposed to predispose males to worse outcomes include increased inflammatory marker production in males compared to females [22], increasing the probability of cytokine storm, linked to the pathogenesis of poor outcomes in COVID-19 infection [23]. The relative biological lack of estrogen and its anti-inflammatory effects in men compared with women, further predisposes male patients to developing cytokine storm with resulting poor outcomes [24].

## Severe COVID-19 and race and ethnicity

Most of our study population were non-Hispanic and Caucasian, and within this homogenous population, there were no differences in outcome when race was taken into consideration. Greater than 90% within each outcome group were Caucasian, and the remaining patients were Black. However, our study showed that a greater proportion of Hispanics were intubated than had a favorable outcome, or underwent in-hospital mortality, and this was a significant finding. Multiple studies have shown a racial and ethnic disparity in COVID 19 deaths, with Black and Hispanic patients experiencing greater mortality than other racial and ethnic groups [25–28]. The possible contributing factors to poorer outcomes in Black and Hispanic patients are likely multifactorial and include factors such as an increased prevalence of comorbidities and behavioral variables that predispose to worse outcomes in COVID-19 infection, such as hypertension, congestive heart failure, chronic lung diseases, and smoking [26,29]. It has also been suggested that socioeconomic disparities limiting access to healthcare and impacting living conditions may also play a part in the disproportionate mortality rate from COVID-19 infection in minorities [26]. Vitamin D deficiency has been postulated as a possible contributing factor to poor outcomes in Black patients. Blacks have been shown to have the highest rate of vitamin D deficiency compared to other racial groups. In a study by Liu et al, the prevalence of vitamin D deficiency was 71.9% in non-Hispanic Black adults in the US, compared to 18.6% in non-Hispanic Caucasians, and 42.8% in Hispanics [30]. Chiodini et al showed that vitamin D deficiency was associated with a 2-times greater odd for ICU admission and mortality in COVID-19 patients [31]. The significance of vitamin D in COVID-19 infection is highlighted by studies that have elaborated the roles played by vitamin D in antiviral immunity and anti-inflammation [32]. These studies highlight the need for targeted primary care and preventative efforts towards minorities who are at increased risk of poor outcomes from COVID-19 infection.

## Severe COVID-19 and age

Although our study highlights a trend for older patients to have worse outcomes, this was not statistically significant, unlike findings from other studies. A Belgian study by Molenberghs

et al shows a stepwise increase in COVID-19 deaths with age, with the greatest number of deaths occurring in patients in their 90s [33]. Another study shows a steep exponential relationship between increasing age and severe, critical COVID-19 infection. For example, around the age of 25 years, fatal disease occurs in about 0.01% of their study population, compared with almost 10% of fatal disease in those aged 75 [34]. Other studies have also had similar findings [35–37]. The increased risk of severe illness in older individuals might be related to factors such as a generally diminished immune system [38], and a specific age-related decline in lung-associated immune function and accumulation of inhaled particulate matter within pulmonary macrophages, leading to weakened phagocytosis [39].

## Severe COVID-19 and vaccination status

In our population of patients with severe COVID-19 infection, the rate of full vaccination was very poor (7.9%), and there was no difference in vaccination rates between outcome groups. Multiple studies have shown a link between a lack of vaccination and severity of COVID-19 infection [40], including hospitalization [41], requirement for invasive ventilation [42,43], and mortality [42]. Our study describes a poor rate of vaccination in a population with severe COVID-19 but does not show a difference between outcome categories.

## Severe COVID-19 and comorbidities

**Body mass index.**   Numerous studies have linked greater BMIs to increased severity of COVID-19 infection, including requirement for invasive ventilation and mortality from disease [44,45], although our study revealed no association with BMI. Some of the suggested pathophysiologic mechanisms underlying the severity of COVID-19 infection in obese individuals include the association of obesity with chronic inflammation, known to hinder immunologic responses to pathogens [46]. Furthermore, it has been suggested that the innate and adaptive immune responses differ in overweight and obese individuals than normal weight individuals; an outcome due to leptin and lower adiponectin concentrations, resulting in a dysregulated immune response [47,48].

Obesity has also been shown to decrease lung function, including decreasing the forced vital capacity, forced expiratory volume, and the overall respiratory system's capacity to function, with a resultant poor baseline respiratory status [49]. Such decreased baseline respiratory status could be exacerbated by COVID-19 infection, resulting in a need for supportive ventilation [50]. Approximately 42.4% of individuals suffer from obesity in the United States [51], indicating that a substantial proportion of our population is at increased risk of severe COVID-19 infection based on BMI alone. Additionally, studies have shown a strong association between high BMIs and an increased risk for developing other chronic illnesses, such as heart disease, type 2 diabetes [52], chronic kidney disease, hypertension, and dyslipidemia [53], further increasing the risk of disease severity by other pathophysiologic mechanisms. Although statistically insignificant, our study reveals a trend between increasing BMI and disease severity. Moreover, some other studies did not find any associations between COVID-19 and BMI [54,55]. Ultimately, obese patients should be continuously monitored to properly manage potential escalation of respiratory support needs.

**Chronic kidney disease.**   One of the most frequent risk factors for severe COVID-19 was CKD, globally [56]. While CKD was prevalent in both the favorable and intermediate groups, it was not in the poor outcome group. This finding differs from other studies, that have shown CKD to be a major risk factor for both severe disease and mortality in COVID-19 infection [57–59].

CKD has emerged as the comorbidity that not only carries the highest risk for severe COVID-19 but also the most common comorbidity, despite the fact this was not initially

noted in medical publications pertaining to risk factors for severe COVID-19 [57]. The poor outcomes seen in CKD patients with COVID-19 is likely multifactorial. Patients with CKD following renal transplants are likely on immunosuppressive medications that dampens their immune response to the virus, predisposing them to more invasive and poorly controlled infections [60]. Furthermore, it has been suggested that worse outcomes in CKD patients may be related to worsened pulmonary inflammation, due to a baseline pro-inflammatory state in CKD, and impaired innate and adaptive immune systems [61,62]. Moreover, other researchers have suggested that CKD, especially at the end-stage renal disease phase, is an acquired immunodeficiency state that predisposes to upper respiratory tract infections, greater than the risks faced by the non-CKD general population [63].

**Type 2 diabetes mellitus.** Our study revealed no statistical association between COVID-19 severity and DM2. While some studies suggest that DM2 does not necessarily increase the risk of getting COVID-19 infection, their findings do indicate that DM2 predisposes to severe COVID-19 infection [64]. More studies have supported the notion that both type 1 and type 2 DM are associated with an increased odds for mortality compared to non-diabetic patients [64–66]. Furthermore, COVID-19 in DM and uncontrolled hyperglycemia has been associated with an increased risk for respiratory and cardiac compromise, and increased risk for ICU admission [66,67]. In non-diabetic patients, elevated blood glucose on hospital admission has been linked with increased mortality risk [68]. COVID-19 infection not only leads to worsening of baseline DM but could also lead to new-onset DM and poor glycemic control [69]. The poor outcomes that have been associated with DM are likely driven by multiple pathophysiologic mechanisms, including insulin resistance resulting from the proinflammatory state caused by COVID-19 infection [70], and impaired immunity including impaired neutrophil chemotaxis and phagocytosis [71]. Both increased ACE2 and increased furin (a type-1-membrane-bound protease) have been associated with DM; both proteins are associated with entry of the SARS-CoV-2 virus and coronaviruses, respectively, into cells [72,73]. It could therefore be deducted that DM predisposes to increased invasion of the respiratory system and worse respiratory outcomes [71]. Furthermore, ACE2 is not only found in the respiratory epithelium, but also in other organs, such as the renal cortex, liver, and the pancreas [72–74]. Invasion of the virus into cells leads to inflammation that could ultimately damage cells. In the case of diabetic patients, damage to the beta cells in the pancreas could further exacerbate underlying DM, and in those without DM, cause uncontrolled hyperglycemia, that further perpetuates a proinflammatory state that contributes to the cytokine storm which is associated with severe infections. These mechanisms would also account for hyperglycemia seen in non-diabetic COVID-19 patients [70,72].

**Hypertension.** According to an analysis of adults hospitalized with COVID-19 between December 2021 and April 2022, hypertension more than doubled the risk for hospitalization from the Omicron-variant of COVID-19 infection, even in the presence of full vaccination, including a booster dose of the COVID-19 vaccines. Our study had a population prevalence of hypertension of 67.4%, and the intermediate group had the greatest prevalence at 77.8%, although this was not statistically different from the favorable or poor groups. Studies have also found that hypertension has the greatest risk for hospitalization when compared with other comorbidities [75]. Another study showed that hypertension, along with diabetes and cardiovascular disease were the most common comorbidities in their COVID-19 population, and hypertension trended in association with disease severity(p = 0.085) [76]. Other studies have shown that hypertension was associated with greater infection severity, and that hypertensive compared with normotensive patients had a greater incidence of ICU admissions, necessity for mechanical ventilation, and mortality [10]. One of the major pathophysiological mechanisms suggested for the association of hypertension with worse outcomes in COVID-

19, is based on the renin-angiotensin-aldosterone system(RAS). This system ultimately leads to elevated blood pressure, increased fibrosis, increased inflammation, and increased production of reactive oxygen species [77]. Renin is produced by the juxtaglomerular cells of the kidneys; renin converts angiotensinogen to angiotensin I(Ang I), which is cleaved into Angiotensin II (Ang II) by angiotensin converting enzyme (ACE) in the lungs. Ang II is the main effector of the RAS system and leads to the effects previously mentioned after binding angiotensin II type 1 receptors ($AT_1R$) [78]. The enzyme, angiotensin converting enzyme 2 (ACE2) acts as a counter regulator of the RAS system [79]. This enzyme cleaves Ang II into angiotensin 1–7, which binds to the Mas receptor and results in increased activity of nitric oxide synthase, vasodilation, anti-inflammatory activity, and anti-fibrosis activity. These effects result in vasodilation, decreased blood pressure, and are ultimately cardioprotective [80]. ACE2 also cleaves a precursor of the RAS system, Ang I, into angiotensin 1–9, which binds to angiotensin II type 2 receptors ($AT_2R$) and results in similar cardioprotective and vasodilatory effects like angiotensin 1–7 [81]. However, $AT_2R$ is sparse in adult human tissues, including cells of the cardiovascular system, which means that the major counter-regulatory mechanism against the RAS pathway within the pathway itself, is via ACE2 [82]. The significance of ACE2 within the pathophysiologic mechanism by which the severity of COVID-19 infection is elucidated in hypertensive patients, is seen by understanding how the SARS-CoV-2 virus enters cells via ACE2 [19]. After the binding of the SARS-CoV-2 virus via its spike protein to ACE2, the virus is internalized into the cells. This is followed by shedding of ACE2 from cell surface membranes, which ultimately results in decreased expression of ACE2 in tissues affected by SARS-CoV-2 virus [83,84]. The lack of a major component (i.e., ACE2) of the counter-regulatory mechanism of the RAS system, leads to poorly controlled blood pressures and pulmonary hypertension. These outcomes contribute to endothelial dysfunction and pulmonary oedema formation, with subsequent worsened pulmonary function and possible ARDS. These mechanisms can result in worse outcomes in COVID-19 infection [85].

**Congestive heart failure.** The prevalence of CHF in our study population was 15.7%. Patients who experienced in-hospital mortality had a greater prevalence(16%) of CHF than those who were intubated but did not experience in-hospital mortality (11.1%); however, neither this finding nor any of the intergroup comparisons were significant Table 2. A metanalysis involving over 150,000 COVID-19 patients from 198 papers showed that CHF was associated with a >11 times odds for mortality; this was greater than the odds associated with hypertension and other cardiovascular diseases [86]. Another metanalysis showed that the relative risk of death in COVID-19 patients with CHF was 3.38(95% confidence interval (CI) 1.80–6.32), indicating an association with severe COVID-19 infection and CHF [87]. Other studies have also shown CHF to be a strong risk for hospital admission, critical illness [14], mechanical ventilation, and mortality [29,88–90]. Several pathophysiologic mechanisms have been suggested to explain poor outcomes seen in CHF patients with COVID-19 infection. One possible mechanism suggested is via myocardial damage, either directly or indirectly. These mechanisms culminate in exacerbation of underlying heart failure, worsening baseline oxygenation status, which in combination with the mechanisms by which the COVID-19 infection worsens oxygenation (i.e., ARDS), increases the requirement for respiratory support [88]. Fever from the infection may result in tachycardia that if sustained, could lead to decreased diastolic filling and stroke volumes, pushing a heart with underlying congestive problems into an acute exacerbation [91]. This resulting tachycardia can also lead to worsening of functional parameters in a failing heart, or to the development of new-onset heart failure via tachyarrhythmia-induced cardiomyopathy [92]. Examples of indirect mechanisms that have been suggested include the inflammatory milieu resulting from COVID-19 infection, which favors endothelial dysfunction, with resulting thrombotic manifestations, which may also include the pulmonary

and coronary vasculature [93–95]. Such pathological mechanisms could lead to poor oxygenation of the myocardium, with resulting myocardial dysfunction that could subsequently push a congestive heart into acute exacerbation, worsening the baseline respiratory status, necessitating respiratory support.

**Chronic lung disease, COPD, OSA.** At the beginning of the COVID-19 pandemic, it was quickly hypothesized and supported by real-world data that patients with underlying lung disease were at increased risk for severe infection [96]. Our study shows a stepwise increase in the proportion of patients with CLD within each subsequent outcome group (19.6%, 27.8%, and 40.0%, respectively) (p = 0.002) Table 2. These findings indicate a statistical association between the severity of COVID-19 infection and underlying CLD. Furthermore, although our study showed that the poor outcome group had the greatest prevalence of COPD(28.0%) compared with the intermediate(11.1%) or favorable (17.4%) groups, this finding was not significant. However, similar to the compiled group of CLD, OSA showed a stepwise increase in prevalence with each subsequent outcome group Table 2. Other studies support our findings, and have specifically shown an increased risk for severe forms of COVID-19 infection in the presence of baseline CLD. Studies have shown that chronic lung disease was associated with an increased odds for requiring hospitalization, intensive care unit level of care, and in-hospital mortality [97–100]. Cade et al. showed an increased all-cause mortality in COVID-19 patients with OSA compared with controls without OSA (OR: 1.79, 95% CI 1.31–2.45; p<0.001) [101]. Similar mortality trends in COVID-19 patients with OSA [9,102], and COPD [103,104] have been shown by other studies. Poor outcomes in patients with OSA and COPD may been attributed to a baseline heightened inflammatory state [105], which is associated with increased oxidative stress and impaired immunity [106]. When this background of inflammation and impaired immunity is combined with the COVID-19 infection, an additive or synergistic effect may result, amplifying the cytokine storm associated with worse respiratory outcomes [107,108]. Another possible mechanism for poor outcomes in OSA patients with COVID-19 infection, is via vitamin D deficiency [108]. One study has shown that vitamin D levels are lower in patients with OSA [109]. Vitamin D is associated with reducing oxidative stress, has anti-inflammatory activities, and is involved in regulating cellular immunity including antiviral immunity [32,110]. A deficiency of vitamin D may predispose to worse outcomes; this is supported by studies in which vitamin D deficiency was associated with ICU admission and mortality in COVID-19 infection [31]. OSA is also associated with elevated levels of angiotensin II and aldosterone [111]. As previously elaborated earlier in this manuscript, ACE2, the membrane-bound enzyme by which the SARS-CoV-2 virus enters cells, is also a counter-regulator of the RAS system. ACE2 cleaves Ang II into angiotensin 1–7, a molecule which results in vasodilation and has anti-inflammatory, antifibrosis and cardioprotective effects [79,80]. It may be suggested that an elevation of Ang II in a setting where ACE2 becomes diminished due to shedding from the cell surface after binding the SARS-CoV-2 virus [83,84], leads to an imbalance of the RAS system, veered towards the Ang II pathway, which may exacerbate systemic and pulmonary hypertension, with resulting cardiotoxic effects and respiratory compromise [85]. There is no single pathophysiological pathway by which chronic lung diseases increase the risks of poor outcomes in COVID-19 infection, but rather multiple mechanisms working in concert.

**Smoking history.** Our study reveals a notable trend in which the poor outcome group had the greatest prevalence of a smoking history; however, there were no differences in outcomes when current versus former smokers were compared Table 2. Other studies show greater mortality in smokers than non-smokers [112–114], and a higher risk of severe disease and mortality in current smokers than in former smokers, unlike our study [115]. The mechanisms suggested to explain this increased risk of poor outcomes, include an increased baseline

inflammatory state in smokers, which increases oxidative stress with resulting immune dysfunction. This combination of increased inflammatory cytokine and immune dysfunction creates a fertile background for poorly controlled invasion and proliferation of the SAR-CoV2 virus [116,117]. In addition to the poor respiratory reserve found in patients with a smoking history, the elaborated pro-inflammatory state created by the COVID-19 infection may lead to ARDS, necessitating respiratory support, and may eventually lead to death [112].

### Hospital course in severe COVID-10

**Markers of liver injury.**    Aminotransferases were most greatly elevated in the poor outcome group in our study. While these findings were significant for aspartate amino transferase (AST), they only trended for alanine aminotransferase(ALT) Table 4. Bloom et al showed that patients who underwent intubation had on average, greater peak AST and ALT values than those who were not intubated [118]. Similar to other studies [118], the pattern of enzyme elevation in our study was consistent with hepatocellular injury. Another study showed that a greater proportion of patients with AST and ALT values greater than the upper limit of normal experienced the primary composite outcomes of ICU admission, mechanical ventilation, and death, than those with normal values [119]. Liu et al., further support the findings that those with severe COVID-19 infection have greater elevations in their AST and ALT levels, compared with those with non-severe disease; however, stratification was not done within the severe group, unlike our study [120]. Another study stratifies patients based ICU hospitalization versus non-ICU hospitalization, and show a greater median ALT in the former than the latter (49.0 U/L vs 27.0 U/L, p = 0.038), with similar findings for AST. (44.0 U/L vs34.0 U/L, p = 0.10). When the proportion of patients with AST<40 was considered, there were more patients in the non-ICU group(75%) than the ICU group(38%), and this was significant (p = 0.025) [121].

These studies indicate a relationship between COVID-19 severity and markers of liver injury, specifically that more severe COVID-19 infection appears to be associated with greater increases in markers of liver injury. One plausible mechanism of liver injury in COVID-19 infection is direct invasion of liver cells by the virus via the ACE2 receptor (ACE2r), similar to invasion in the respiratory epithelium. RNA sequencing studies have shown that hepatocytes express ACE2rs [122], however poorly so (about 3% expression), compared to cholangiocytes with almost 60% expression [123]. While Chai et al have suggested that liver injury may occur via cholangiocyte injury, a specific mechanism was not suggested [123]. Both murine studies and human tissue studies have depicted an increase in ACE2r expression following liver injury and cirrhosis [124,125]. Findings suggest that the upregulation of ACE2r was secondary to compensatory proliferation of hepatocytes that were derived from bile duct epithelial cells [126]. It could be deducted that an initial liver insult results in upregulation of ACE2r, which may subsequently create an environment that supports greater invasion and pronounced hepatocyte injury by the SARVS-CoV2 virus, although such a hypothesis has not been proven. Nonetheless, given the low expression of ACE2rs in liver cells, the origin of the initial insult in COVID-19 infection, if such a hypothesis is to be considered, must be postulated.

Besides the ACE2r, it has been shown that the SARS-CoV-2 virus is also able to invade cells via different receptors, specifically the dendritic cell-specific intercellular adhesion molecule-grabbing nonintegrin(DC-SIGN), which is a transmembrane adhesion molecule found on dendritic cells [127,128]. SARS-CoV-2 along with other viruses including HIV-1 and CMV can enter dendritic cells via this molecule. The virus is subsequently hypothesized to avoid antigen processing within dendritic cells, but instead uses these cells as a transportation route to tissues where they can invade and institute an infection [127,128]. Similar transmembrane

molecules have been described in human liver cells and are known as liver/lymph node-specific intercellular adhesion molecule-grabbing nonintegrin(L-SIGN). These liver transmembrane molecules can equally facilitate the entry of the virus into hepatocytes [128,129]. Although it has not been shown that the virus can replicate after entry via L-SIGN, it is reasonable to assume that an initial injury may occur to hepatocytes via this route, with subsequent ACE2r upregulation that exacerbates liver injury. The differential response in liver injury in patients may be associated with genetic differences and polymorphisms in the aforementioned receptors. Studies have suggested that liver injury occurring in the presence of COVID-19 infection may likely be caused by cytokine storm syndrome(CSS) and drug-induced liver injury. CSS is a state in which an infectious or non-infectious insult leads to the proliferation of immune cells of the innate and adaptive systems, with subsequent elaboration of cytokines that mediate increased inflammation with associated organ dysfunction. It has been suggested that slight genetic polymorphisms in receptors involved in cytokine function and effect may be responsible for differential manifestations of CSS, or the lack thereof, in patients exposed to similar insulting stimuli [130]. In CSS, the resulting immune cells attack tissue cells leading to apoptosis and necrosis, with resulting organ dysfunction and an exacerbated inflammatory process [131]. Hypoxia-induced liver injury is also one of the mechanisms that has been postulated, given as COVID-19, especially severe disease, leads to acute hypoxic respiratory failure. Hepatocyte hypoxia further exacerbates inflammation that perpetuates liver damage [131,132]. The mechanisms of liver injury in COVID-19 infection appear to be multifactorial, and more severe disease appears to be associated with greater elevations in liver enzymes.

**AKI.** Our study shows a progressive increase in the prevalence of AKI in each subsequent outcome group, with a general population incidence of 9%. AKI occurred in 4.3%, 5.6% and 20.0% of each subsequent outcome group respectively, although these findings were not significant. Cheng et al also showed an association between AKI and in-hospital mortality [62], as did Richardson et al [7]. Other studies have shown that AKI was more prevalent in patients who required ICU care than those who did not [2]. One possible mechanism suggested for the occurrence of AKI in COVID-19 infection involves the attenuation of the ACE2 receptors (ACE2rs) in the kidneys. Studies have shown that ACE2rs are present in the kidneys [133], and that SARS-CoV-2 is capable of invading renal cells [134–138]. As previously explained in this manuscript, SARS-CoV-2 enters cells via ACE2rs; binding to this receptor by the viral spike protein leads to endocytosis of the ACE2r with resultant decreased expression on cell surfaces. ACE2 cleaves Ag II into angiotensin 1–7, acting as a counter-regulator of the pro-inflammatory and pro-thrombotic effects of Ang II. It may be proposed that an attenuation of ACE2 after binding the virus in the kidneys may also institute similar inflammation and thrombotic complications similar to what occurs in the lungs, due to uninhibited Ang II, leading to kidney injury [139,140]. In addition to cleaving Ang II into angiotensin 1–7, ACE2 is also involved in cleaving an active product of the kinin-kallikrein system(KKS) [141]. The KKS involves the cleavage of high molecular weight kininogen into bradykinin and des-arg-973-bradykinin(DABK). Bradykinin is involved in causing vasodilation and decreasing blood pressure. DABK has been shown to be involved in promoting inflammation, including increasing capillary permeability, vasodilation, and leukocyte recruitment [142]. ACE2 breaks down DABK and therefore acts as a counter regulator of the DABK pathway [143]. In a situation when ACE2 is diminished, DABK can promote inflammation in a less inhibited manner. It can therefore be inferred that increased DABK in the pre-renal vasculature may lead to decreased renal perfusion and subsequent ischemic renal injury, which could lead to AKI. Similarly, within the renal vasculature, increased DABK may promote inflammation within the kidneys, which could also lead to AKI [139].

It has also been suggested that direct invasion of renal cells by the virus may lead to tubular cell damage with resulting acute tubular necrosis and AKI [139]. Other hypothesized mechanisms of renal injury in COVID-19 infection include the cytokine storm-mediated septic shock, with resultant renal hypoperfusion and hypoxia, which could lead to AKI [139]. Furthermore, hypoxia resulting from shock may also lead to non-traumatic rhabdomyolysis, as could direct invasion of skeletal muscles by the virus; these mechanisms could cause acute tubular necrosis [139]. Similar to other pathological consequences of COVID-19 infection, the mechanisms by which the virus is proposed to cause kidney injury are myriad.

**Severe COVID-19 infection and respiratory parameters.** Arterial blood gases and respiratory indices are important to monitor patient status, determine need for assisted ventilation, and predict outcomes in COVID-19. Our results show that differences in average $PaCO_2$, $PaO_2$ and pH are associated with prognostic outcomes.

The average $PaCO_2$ for the entire study population was within normal limits at 41.4mmHg; however, only the poor outcome group had an average $PaCO_2$ level that was greater than the upper limit of normal (46.4mmHg). In a similar manner, the poor outcome group had the highest maximum $PaCO_2$ compared to the other outcome groups. The inflammation resulting from COVID-19 infection leads to increased vascular permeability in the pulmonary vasculature, increasing the alveolar-capillary membrane, resulting in decreased gas diffusion across this membrane and subsequent $CO_2$ retention and poor oxygen uptake [144]. It follows that more severe infection would lead to greater extravasation from the pulmonary vasculature, greater pulmonary edema, and a greater barrier to diffusion of gases [144]. Studies have also shown abnormal shunting of pulmonary perfusion to poorly ventilated lung areas, exacerbating hypoxia in the already inflamed lungs [145]. Studies have indicated that hypercapnia early in the course of admission might be indicative of severe ARDS, requiring ICU care [146]. Other studies have linked hypercapnia to the occurrence of thromboembolic events, longer mechanical ventilation, ICU stay, and longer length of hospital stay. Furthermore, hypercapnia has been associated with comorbidities including COPD and increased BMI above normal limits; however, mortality was not increased in hypercapnic, compared with normocapnic patients [146,147]. These results differ from our study regarding mortality, in that the group that experienced in-hospital mortality had the greatest average and maximum $PaCO_2$ than the other outcome groups, and this was a significant finding. The comorbidities associated with hypercapnia, including COPD and obesity, have been discussed earlier in this manuscript. It has been suggested that hypercapnia has both beneficial effects, such as reduction of oxidative stress and promoting oxygenation, and also deleterious effects; the net result effect depends on the balance of effects [148]. Such a hypothesis may explain why certain studies show a net non-mortality effect, while others like ours show a statistical association with mortality [149,150]. Nonetheless, it is unclear which factors drive the balance in either direction.

Moreover, our results show a statistically significant association between lower blood pH and worse outcomes Table 5. These findings agree with previous studies [151]. Lower recorded pH values may be secondary to carbon dioxide accumulation, which is also shown to be greatest in the poor outcome/in-hospital mortality group in our study Table 5. Acidosis may also be associated with worse inflammation with resulting tissue hypoxia and lactic acidosis, all of which are associated with poorer outcomes in severe COVID-19 infection [152]. Although our study shows no statistical association between average oxygen saturation on admission and outcome groups, other studies have associated lower admission oxygen saturation with poor outcomes which include increased mortality [153], and increased requirement for ICU care [154]. These findings suggest that admission oxygen saturation may be a predictor of ICU level care during admission, although our study did not make such a distinction [154].

## Medications

**Hydroxychloroquine.** Approximately 4–5% of patients in each outcome group received hydroxychloroquine during admission, and there were no significant differences between the outcome groups and hydroxychloroquine exposure in our study. In a randomized trial of 491 patients, at 14 days, there were no differences in symptoms experienced by patients in the hydroxychloroquine group versus the placebo group. While 24% of participants receiving hydroxychloroquine had ongoing symptoms, 30% receiving placebo experienced symptoms (p = 0.21). This trial concluded that hydroxychloroquine did not substantially reduce symptom severity in outpatients with early, mild COVID-19 [155]. Another retrospective study involving data collected from 6 continents including North and South America, Europe, Africa, Asia, and Australia, showed that hydroxychloroquine was administered more often in the group that experienced in-hospital mortality(5.1%), than in the survival group(2.9%); hazard ratio(HR):1.335, 95% CI 1.223–1.457 [156]. Multiple studies have provided contradicting results regarding the efficacy of hydroxychloroquine in COVID-19 infection and outcomes, with some studies supporting efficacy [157,158], and others showing a lack of efficacy and increased adverse effects with hydroxychloroquine regimens [159–162]. The mechanisms suggested to be involved in the antiviral activities of hydroxychloroquine include its ability to decrease the elaboration of pro-inflammatory cytokines, by inhibiting antigen presentation by antigen presenting cells, including B cells. Another mechanism involves the glycosylation of the ACE2 receptors, which inhibits the binding of the viral spike protein to the receptor, thereby inhibiting viral invasion [163,164]. There is currently not enough evidence to support, or definitively refute the efficacy of hydroxychloroquine in COVID-19 infection.

**Tocilizumab.** In our study, approximately 30% of patients in each subgroup received tocilizumab, and there were no statistical differences between subgroups. Another retrospective study of 1,938 patients with confirmed SARS-CoV-2 pneumonia found that patients who received tocilizumab experienced more inpatient mortality(26.1% vs 13.2%), higher ICU admission rates(56.4% vs. 20.7%), and increased rate of mechanical ventilation (33.6% vs. 11.4%), compared with the patient population that did not receive tocilizumab treatment. This study concluded that tocilizumab did not improve inpatient mortality rate for COVID-19 patients [165]. Martinez-Sanz et al., showed similar results, in that ICU admission (19% vs 3%), overall mortality (23% vs 12%), and ICU mortality (32% vs 19%) was greater in the group that received tocilizumab. Tocilizumab was associated with a higher risk of death in the initial analysis (HR 1.53, 95% CI 1.20–1.96, p = 0.001). However, when their study cohort was stratified by CRP levels (<150 vs >150), tocilizumab was associated with a decreased rate of death than in the population who did not receive tocilizumab (HR 0.34, 95% CI 0.17–0.71, p = 0.005) [166]. Studies showing a positive association between tocilizumab and ICU admission or mortality, may indicate that a high-risk group of patients were being selected; specifically, these patients had severe disease that was not responding to standard of care treatment, necessitating the addition of supplemental therapy. Other studies have shown tocilizumab to be associated with improved survival [167–170]. The most up to date consensus appears to suggest that tocilizumab is beneficial in reducing mortality; however, in our study that selected severe COVID-19 patients, tocilizumab was similarly administered in all outcome groups, and there was no statistical association with mortality.

**Baricitinib.** In our study, baricitinib was statistically associated with the poor outcome group Table 6 in S1 Table. According to a meta-analysis of 52 studies of which 4 studies were randomized controlled trials, baricitinib significantly reduced mortality and disease progression in the patient population treated with a dosage of 2 mg or 4 mg, for a maximum duration of 14 days of baricitinib [171]. These findings are supported by another meta-analysis in which

a pooled analysis showed a statistically significant reduction in 28-day mortality in the baricitinib arm, compared to the standard of care arm (OR 0.69, 95% CI 0.50–0.95, p = 0.04) [172]. Despite this portrayed improved mortality, progression to respiratory failure requiring positive pressure ventilation, and progression to invasive ventilation or extracorporeal membrane oxygenation appeared to be greater in the baricitinib group than the standard of care group [172]. In another non-controlled retrospective cohort study on moderate-to-severe infection, baricitinib was temporally associated with clinical improvement and recovery [173]. The mechanism of action behind the improved outcomes with baricitinib administration is associated with the reduction of the cytokine storm associated with severe cases of COVID-19 infection [174,175]. Although our study showed a statistical association between baricitinib and poor outcomes, specifically in-hospital mortality, this might be an effect of late administration in patients in whom outcomes would have been poor regardless of the drug administered, although we have not explored such an effect in this study.

**Azithromycin.** Although a greater proportion of patients in each subsequent outcome group was administered azithromycin, there were no statistical differences(p = 0.268), Table 6 in S1 Table. Azithromycin is a broad-spectrum macrolide antibiotic with anti-inflammatory properties, which appear to be the basis for consideration in COVID-19 treatment [176,177]. Some studies have also indicated in-vitro antiviral activity [178], including against RNA viruses such as Zika, rhinovirus and SARS-CoV-2 [179,180]. Despite these proposed beneficial effects of azithromycin in-vitro, real life studies have not shown such effects. A randomize trial by Furtado et al concluded that azithromycin added to standard of care treatment did not have any beneficial effects including on mortality, need for ventilation, or other clinical status [181]. Similarly, Cavalcanti et al did not show any improved outcomes when azithromycin was added to hydroxychloroquine and compared against standard of care. Rather, adverse events appeared to be more prevalent in the azithromycin group [162]. Other studies also support the notion that azithromycin neither results in clinical improvement nor reduce the risk of poor outcomes [182–184], similar to the findings in our study.

**Dexamethasone.** Patients in the poor outcome group had a significantly longer average number of days on dexamethasone than the other outcome groups (p = 0.002) Table 6 in S1 Table. Although during the early phases of the pandemic the use of glucocorticoids was controversial [185], dexamethasone has now become a staple in the standard of care treatment for COVID-19 infection, especially in moderate-to-severe infections, owing to multiple studies supporting improved mortality and clinical outcomes when glucocorticoids are administered [186–192].

Although Chaudhuri et al have suggested that longer duration of low-dose glucocorticoids is associated with improved survival [193], our study shows a clear and statistically significant pattern in which a greater number of days on high-dose dexamethasone was associated with worse outcomes. Whether this finding might have arisen due to the inherent detrimental effects of prolonged glucocorticoid exposure, or prolongation of treatment in very sick patients that showed no signs of improvement, is unclear and not explored by our study. However, studies have indicated that prolonged glucocorticoid therapy [194], and late administration during the course of severe COVID-19 infection [195] may be associated with increased mortality. Autopsy findings have confirmed the prothrombotic state involved in COVID-19 infection; specifically, autopsy studies have shown large vessel thrombi and microthrombi in infected patients [196]. Glucocorticoids have also been shown to have prothrombotic effects by increasing clotting factors and fibrinogen concentration [197,198]. The combination of a prothrombotic inflammatory state created by COVID-19 infection, and prolonged exposure to prothrombotic glucocorticoids, may exacerbated microthrombi and large vessel thrombi

leading to worse outcomes [199]. Therefore, the duration of glucocorticoid treatment in severe COVID-19 infection may be relevant to outcomes, as has been indicated by our study.

**Angiotensin converting enzyme inhibitors(ACEi)/Angiotensin receptor blockers (ARB).** While patients who experienced in-hospital mortality had the smallest prevalence of ARB and ACEi use during admission, this finding was not statistically significant Table 6 in S1 Table. Some studies have suggested that ACE inhibition leads to a lack of Ang II, which in turn upregulates ACE2 receptors (ACE2r), which ultimately promotes anti-inflammation and anti-fibrinolysis via Angiotensin 1–7, as previously discussed in this manuscript [200]. Other authors have suggested disruption of the RAS pathway by ACEi, or ARBs leads to a disruption of the ACE2/Angiotensin 1-7/MAS pathway, with resultant decreased ACE2, and subsequently decreased viral entry into cells [201]. Such a hypothesis is supported by murine models which showed that ACE2 knockout mice had significantly lower viral replication and less lung injury after infection with SARS-CoV-1, compared with wild type mice [85]. Feng et al showed that administration of ACEi/ARBs as antihypertensives was more prevalent in moderate disease than severe and critical disease [202]. Other studies also support the greater prevalence of ACEi/ARBs administration in less severe disease including less inflammation and lower mortality [203–206]. In contrast, a meta-analysis of 53 studies showed no association between mortality and COVID-19 severity, and ACEi/ARB use [207]. There appears to be no general consensus regarding the use of ACEi/ARB in setting of COVID-19; it is therefore the job of the clinician to weigh the risks and benefits while taking into consideration underlying comorbidities, acute organ dysfunctions, and hemodynamic status, to determine whether or not the use of ACEi/ARBs during admission for severe COVID-19 disease is imperative.

## Study limitations

Our study was limited by several factors. First, our population was very homogenous with regard to race and ethnicity. While other studies have shown a racial and ethnic distribution regarding COVID outcomes, we were not able to test similar findings in our population, owing to a greater than 90% Caucasian make-up of our population. Furthermore, our study was limited to a specific geographic location within the Geisinger health system; COVID-19 outcomes might differ based on geography within the US and this is an aspect of the infection that we were unable to test. Our sample size was small, especially after stratifying patients into 3 separate outcome groups. Some trends according to outcome groups were shown by our study but a level of statistical significance was not reached for some variables.

## Conclusion

Our study showed statistical associations between specific outcome groups in severe COVID-19 infection and clinical variables. In a homogenous study population with severe COVID-19 infection, in-hospital mortality was more prevalent in males than females. Chronic lung disease, obstructive sleep apnea and a history of smoking were associated with in-hospital mortality. Bacterial pneumonia was associated with invasive mechanical ventilation. Average albumin was lowest in patients who were never invasively ventilated. Average AST was greatest in patients who experienced in-hospital mortality. Although average ALT was also greatest in the in-hospital mortality group, this finding was statistically insignificant. Average bilirubin, average lactate, average $PaCO_2$ and maximum $PaCO_2$ were greatest in the in-hospital mortality group. Average CRP was greatest in the combined group of patients who underwent invasive mechanical ventilation and in-hospital mortality. Although AKI was more prevalent in the in-hospital mortality group, this was not a significant finding. The in-hospital mortality group had the most acidotic average pH compared to the other 2 outcome groups. Baricitinib

administration was most prevalent in the in-hospital mortality group, and the average number of days on dexamethasone was greatest in this group. Patients who were intubated with a good (i.e., non-mortality) outcome were intubated earlier during their hospital course, with an average difference of 135.6 hours, when compared with intubated patients with poor outcomes.

## Supporting information

**S1 Table. Post-hoc statistical power analysis.**
(XLSX)

## Author Contributions

**Conceptualization:** Ogheneyoma Akpoviroro, Jameson Woodard.

**Data curation:** Ogheneyoma Akpoviroro.

**Formal analysis:** Ogheneyoma Akpoviroro, Nathan Kyle Sauers.

**Investigation:** Ogheneyoma Akpoviroro.

**Methodology:** Ogheneyoma Akpoviroro.

**Project administration:** Ogheneyoma Akpoviroro.

**Resources:** Ogheneyoma Akpoviroro.

**Software:** Ogheneyoma Akpoviroro, Nathan Kyle Sauers.

**Supervision:** Ogheneyoma Akpoviroro, Jameson Woodard.

**Validation:** Ogheneyoma Akpoviroro, Queeneth Uwandu.

**Visualization:** Ogheneyoma Akpoviroro.

**Writing – original draft:** Ogheneyoma Akpoviroro, Nathan Kyle Sauers, Queeneth Uwandu, Myriam Castagne, Oghenetejiro Princess Akpoviroro, Sara Humayun, Wasique Mirza, Jameson Woodard.

**Writing – review & editing:** Ogheneyoma Akpoviroro.

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
