## [Decision Letter · Decision Letter 0]

2 Oct 2023

PONE-D-23-21699Severe COVID-19 Infection: An Institutional Review and Literature ReviewPLOS ONE

Dear Dr. Akpoviroro,

Thank you for submitting your manuscript to PLOS ONE. After careful consideration, we feel that it has merit but does not fully meet PLOS ONE’s publication criteria as it currently stands. Therefore, we invite you to submit a revised version of the manuscript that addresses the points raised during the review process. Please revise.

We look forward to receiving your revised manuscript.

Kind regards,

Academic Editor

PLOS ONE

Journal Requirements:

4. Please include your tables as part of your main manuscript and remove the individual files. Please note that supplementary tables (should remain/ be uploaded) as separate "supporting information" files. 

Reviewers' comments:

Reviewer's Responses to Questions

**Comments to the Author**

1. Is the manuscript technically sound, and do the data support the conclusions?

Reviewer #1: Yes

Reviewer #2: Yes

2. Has the statistical analysis been performed appropriately and rigorously? 

Reviewer #1: Yes

Reviewer #2: No

3. Have the authors made all data underlying the findings in their manuscript fully available?

Reviewer #1: Yes

Reviewer #2: Yes

4. Is the manuscript presented in an intelligible fashion and written in standard English?

Reviewer #1: Yes

Reviewer #2: Yes

5. Review Comments to the Author

Reviewer #1: Congratulations on a work well done.

Four years after the pandemic, most of the data listed is already known and published everywhere regarding outcomes I different demographics.

However, the way the results were listed and discussed is very appealing to read. Hence, my suggestion to change the way the paper was written and focus on reformatting the manuscript

Reviewer #2: This is an interesting paper by Akpoviroro et al. describing patients with severe COVID-19.

Some comments for the authors' consideration:

1) The study would greatly benefit from a flow chart figure that outlines eligibility and inclusion.

2) The statistical analysis is more descriptive rather than forming associations. Please revise all indications that associations were found. Statistical methods used only describe significant differences between groups.

3) While the title suggests that it is also a literature review, there is no formal literature search conducted or included in the methods

4) The discussion is rather long, and perhaps should be a separate paper itself.

5) Please justify the sample size or include a sample size calculation.

6. PLOS authors have the option to publish the peer review history of their article (what does this mean?). If published, this will include your full peer review and any attached files.

Reviewer #1: **Yes: **Mahmoud Elfiky

Reviewer #2: No

---

## [Author Response · Author response to Decision Letter 0]

20 Oct 2023

1) Thank you for your feedback. We have incorporated a flow chart outlining the inclusion criteria and outcome groups into the manuscript as a figure.

2) Thank you for your thoughtful comment regarding the use of descriptive statistics in our study. We employed the Chi-squared test, also known as the chi-square test for association or contingency, to analyze our data. The primary purpose of the chi-square test in statistical analyses is to demonstrate associations between categorical data. It is important to note that association does not imply causation, and we have taken great care not to suggest a cause-and-effect relationship in our findings. We believe that the term "association" accurately characterizes the outcomes based on the statistical tests conducted in our study. Nevertheless, we genuinely value your input and in response, we have taken the time to clarify and explicitly state that any associations found in our study results are purely statistical in nature. This precautionary measure is to prevent any potential misunderstandings among our readers and to ensure that our work is accurately represented. If we have erroneously indicated a cause-and-effect relationship in any of our study results, please definitely indicate where these errors are, and we will enthusiastically make the appropriate adjustments. Once again, thank you for your insightful comment; we truly appreciate your engagement with our work and helping to enhance the clarity of our research. 

3) Thank you for bringing this to our attention. We acknowledge that we did not explicitly state in the title or elsewhere in the manuscript that a systematic review was conducted. While we conducted a thorough search for existing literature, it was not done in line with a conventional systematic review process, which was not the primary focus of our manuscript. To ensure transparency and avoid any confusion, we have revised the manuscript's title, replacing the term "review" with "overview." We trust that this modification better communicates the nature of the discussion in our manuscript. 

4) Thank you for your feedback on the manuscript length. We appreciate your perspective on this matter and understand your concerns. We have carefully considered your comments alongside our objectives and have decided to maintain the current length of the manuscript. We believe this length is necessary to comprehensively cover the topic and provide valuable insights to our readers. Your input is valuable, and we hope you understand our reasoning. If you have any further suggestions or specific areas you feel could be condensed, we would be more than happy to hear them. 

5) A post-hoc statistical power analysis was calculated using G*Power Calculator v3.1.9.7 provided by Heinrich Heine University Düsseldorf. This was used to determine the likelihood of detection given our sample size. Of the statistically significant variables, it was found that there was a >80% statistical power on 20/28 tests, or 71.4% of significant tests. The eight tests that were significant but not >80% power all had >60% power with the exception of average bilirubin (mg/dL) and ethnicity. These lower powers are most likely due to the small sample size and the disparity of sizes between groups. A supplemental file has been attached detailing these findings.

---

## [Decision Letter · Decision Letter 1]

1 Dec 2023

PONE-D-23-21699R1Severe COVID-19 Infection: An Institutional Review and Literature OverviewPLOS ONE

Dear Dr. Akpoviroro,

Thank you for submitting your manuscript to PLOS ONE. After careful consideration, we feel that it has merit but does not fully meet PLOS ONE’s publication criteria as it currently stands. Therefore, we invite you to submit a revised version of the manuscript that addresses the points raised during the review process.

Please revise.

We look forward to receiving your revised manuscript.

Kind regards,

Academic Editor

PLOS ONE

Reviewers' comments:

Reviewer's Responses to Questions

**Comments to the Author**

1. If the authors have adequately addressed your comments raised in a previous round of review and you feel that this manuscript is now acceptable for publication, you may indicate that here to bypass the “Comments to the Author” section, enter your conflict of interest statement in the “Confidential to Editor” section, and submit your "Accept" recommendation.

Reviewer #1: All comments have been addressed

Reviewer #3: (No Response)

2. Is the manuscript technically sound, and do the data support the conclusions?

Reviewer #1: Yes

Reviewer #3: Yes

3. Has the statistical analysis been performed appropriately and rigorously? 

Reviewer #1: Yes

Reviewer #3: Yes

4. Have the authors made all data underlying the findings in their manuscript fully available?

Reviewer #1: Yes

Reviewer #3: Yes

5. Is the manuscript presented in an intelligible fashion and written in standard English?

Reviewer #1: Yes

Reviewer #3: Yes

6. Review Comments to the Author

Reviewer #1: Congratulations on the work and the paper. It seems everything has been addressed to the slightest detail

Reviewer #3: Congratulations to address COVID-19 risk factors and the importance of being vaccinated.

Your study is well written but there is too much information. Despite being a observational study ,the authors conduct a detailed review of all the variables shown in the results section, giving the impression for the reader that we are reading a narrative review. It is important to adequate the lenght of the writing to not become tiring to the reader. The discussion section should point out the main topics to guide the reader through the authors point of view to support their findings.

I suggest to reduce the discussion section as the core of the paper is a observational study and their findings.

I suggest put table 6 ( medication ) in supplemental matherial and point out in the results and discussion section the most relevant points in this section.

I suggest put the limitations and strenghts section before the conclusion .

7. PLOS authors have the option to publish the peer review history of their article (what does this mean?). If published, this will include your full peer review and any attached files.

Reviewer #1: **Yes: **Mahmoud Elfiky

Reviewer #3: **Yes: **Marcelo Rodrigues Bacci

---

## [Author Response · Author response to Decision Letter 1]

23 Dec 2023

December 23, 2023,

Editor in Chief

PLOS ONE

Subject: Submission of revised manuscript (PONE-D-23-21699) 

Dear Dr. Chen,

We are pleased to re-submit a revised version of our manuscript titled, “Severe COVID-19 Infection: An Institutional Review and Literature Overview”.

Thank you for the opportunity to revise and resubmit this manuscript. According to the deadline provided, I am submitting this revision before 01/15/2024. We would like to thank you and the reviewers for the time taken to review our manuscript, and the insightful recommendations that were provided. We have incorporated the suggested changes into the manuscript as much as possible. We look forward to working with you and the reviewers to move this manuscript closer to publication in PLOS ONE. 

We have tracked any changes made to the manuscript document. We have specifically responded to each reviewer’s comments below. We have numbered each reviewer’s comment, followed with a response detailing how we have addressed the comment. 

Review Comments to the Author

Reviewer #1: Congratulations on the work and the paper. It seems everything has been addressed to the slightest detail

Response to review #1

Thank you for your thoughtful review. We appreciate your time and effort in evaluating our work. We are delighted to hear that you found our paper comprehensive and that we have successfully addressed the details to your satisfaction.

Reviewer #3: Congratulations to address COVID-19 risk factors and the importance of being vaccinated. 

Your study is well written but there is too much information. Despite being a observational study ,the authors conduct a detailed review of all the variables shown in the results section, giving the impression for the reader that we are reading a narrative review. It is important to adequate the lenght of the writing to not become tiring to the reader. The discussion section should point out the main topics to guide the reader through the authors point of view to support their findings. I suggest to reduce the discussion section as the core of the paper is a observational study and their findings. I suggest put table 6 ( medication ) in supplemental matherial and point out in the results and discussion section the most relevant points in this section. I suggest put the limitations and trengths section before the conclusion .

Response to reviewer #3

Thank you for your thoughtful comments. We appreciate your recognition of the detailed overview that all the authors have carefully worked together to collate. 

Our aim is to present an unbiased portrayal of our findings, including both supportive and opposing studies. We have also provided possible explanations for discrepancies and believe our comprehensive approach offers valuable insights in a neutral manner. We value your input and the time dedicated to enhancing our paper. We have significantly condensed the discussion by eliminating less relevant findings, removing over two thousand words and 28 references. Additionally, we have placed Table 6 in the supplemental material file, referencing it appropriately in the main text, and reorganized the limitation and conclusion sections as suggested. We hope that these changes improve the manuscript’s readability while maintaining a thorough coverage of this topic.

Sincerely, 

Ogheneyoma Akpoviroro, MD 

Geisinger Northeast/Geisinger Wyoming Valley Medical Center 

1000 East Mountain Boulevard 

Wilkes Barre, Pennsylvania 18711 

United States 

570-808-3746 

oakpoviroro@geisinger.edu

o.akpoviroro@gmail.com

---

## [Decision Letter · Decision Letter 2]

22 May 2024

Severe COVID-19 Infection: An Institutional Review and Literature Overview

PONE-D-23-21699R2

Dear Dr. Akpoviroro,

We’re pleased to inform you that your manuscript has been judged scientifically suitable for publication and will be formally accepted for publication once it meets all outstanding technical requirements.

Kind regards,

Academic Editor

PLOS ONE

Additional Editor Comments (optional):

Reviewers' comments:

Reviewer's Responses to Questions

**Comments to the Author**

1. If the authors have adequately addressed your comments raised in a previous round of review and you feel that this manuscript is now acceptable for publication, you may indicate that here to bypass the “Comments to the Author” section, enter your conflict of interest statement in the “Confidential to Editor” section, and submit your "Accept" recommendation.

Reviewer #1: All comments have been addressed

Reviewer #4: All comments have been addressed

2. Is the manuscript technically sound, and do the data support the conclusions?

Reviewer #1: Yes

Reviewer #4: Yes

3. Has the statistical analysis been performed appropriately and rigorously? 

Reviewer #1: Yes

Reviewer #4: Yes

4. Have the authors made all data underlying the findings in their manuscript fully available?

Reviewer #1: Yes

Reviewer #4: Yes

5. Is the manuscript presented in an intelligible fashion and written in standard English?

Reviewer #1: Yes

Reviewer #4: Yes

6. Review Comments to the Author

Reviewer #1: Excellent manuscript and all comments have been addressed. Thank you for addressing them all and responding to the reviewers

Reviewer #4: It seems that all the recommendations made by the reviewers were followed in a detailed and careful manner. Congratulations on the work.

7. PLOS authors have the option to publish the peer review history of their article (what does this mean?). If published, this will include your full peer review and any attached files.

Reviewer #1: **Yes: **Mahmoud Elfiky

Reviewer #4: No

---

## [Editor Report · Acceptance letter]

9 Jul 2024

PONE-D-23-21699R2 

PLOS ONE

Dear Dr. Akpoviroro, 

I'm pleased to inform you that your manuscript has been deemed suitable for publication in PLOS ONE. Congratulations! Your manuscript is now being handed over to our production team.

Kind regards, 

on behalf of

Dr. Robert Jeenchen Chen 

Academic Editor

PLOS ONE